# Cardiovascular Disease in Obstructive Sleep Apnea: Putative Contributions of Mineralocorticoid Receptors

**DOI:** 10.3390/ijms24032245

**Published:** 2023-01-23

**Authors:** Mohammad Badran, Shawn B. Bender, David Gozal

**Affiliations:** 1Department of Child Health and Child Health Research Institute, School of Medicine, University of Missouri, Columbia, MO 65211, USA; 2Dalton Cardiovascular Research Center, University of Missouri, Columbia, MO 65211, USA; 3Department of Biomedical Sciences, University of Missouri, Columbia, MO 65211, USA; 4Research Service, Harry S. Truman Memorial Veterans Hospital, Columbia, MO 65201, USA; 5Department of Medical Pharmacology and Physiology, School of Medicine, University of Missouri, Columbia, MO 65211, USA

**Keywords:** obstructive sleep apnea, intermittent hypoxia, mineralocorticoid receptor, aldosterone, cardiovascular disease

## Abstract

Obstructive sleep apnea (OSA) is a chronic and highly prevalent condition that is associated with oxidative stress, inflammation, and fibrosis, leading to endothelial dysfunction, arterial stiffness, and vascular insulin resistance, resulting in increased cardiovascular disease and overall mortality rates. To date, OSA remains vastly underdiagnosed and undertreated, with conventional treatments yielding relatively discouraging results for improving cardiovascular outcomes in OSA patients. As such, a better mechanistic understanding of OSA-associated cardiovascular disease (CVD) and the development of novel adjuvant therapeutic targets are critically needed. It is well-established that inappropriate mineralocorticoid receptor (MR) activation in cardiovascular tissues plays a causal role in a multitude of CVD states. Clinical studies and experimental models of OSA lead to increased secretion of the MR ligand aldosterone and excessive MR activation. Furthermore, MR activation has been associated with worsened OSA prognosis. Despite these documented relationships, there have been no studies exploring the causal involvement of MR signaling in OSA-associated CVD. Further, scarce clinical studies have exclusively assessed the beneficial role of MR antagonists for the treatment of systemic hypertension commonly associated with OSA. Here, we provide a comprehensive overview of overlapping mechanistic pathways recruited in the context of MR activation- and OSA-induced CVD and propose MR-targeted therapy as a potential avenue to abrogate the deleterious cardiovascular consequences of OSA.

## 1. Introduction

Obstructive sleep apnea is a chronic and highly prevalent condition estimated to affect close to 1 billion people worldwide [1]. OSA is characterized by intermittent increases in upper airway resistance in the context of heightened airway collapsibility, ultimately leading to either reductions or altogether cessation of airflow (i.e., hypopneas and apneas, respectively). These events, which usually last 10–30 s, but can last minutes in severe cases [2], lead to important physiological alterations such as intermittent hypoxia (IH), sleep fragmentation (SF), intermittent hypercapnia, and enhanced intrathoracic pressure swings [3,4,5,6,7]. These processes can, in turn, activate and propagate a large array of pathophysiological mechanisms, inducing sustained enhancements of sympathetic activity coupled with parasympathetic withdrawal, disturbances of the hypothalamic–pituitary–adrenal axis, activation of the renin–angiotensin–aldosterone system (RAAS), systemic oxidative stress, both diffuse and localized inflammation and immune dysregulation, and overall hemodynamic stress. As a consequence of these processes, adverse consequences such as endothelial dysfunction, arterial stiffness, vascular insulin resistance, and cardiac remodeling, ultimately causing hypertension and atherosclerosis, manifest over time [8,9,10,11,12,13,14,15,16,17,18,19,20,21,22,23,24,25,26,27,28,29]. Predictably, OSA has consistently emerged as an independent risk factor for cardiovascular disease (CVD), including coronary artery disease (CAD), stroke, myocardial infarction (MI), congestive heart failure (HF), and several types of arrhythmias, such as atrial fibrillation (AF) [4,6,30,31,32,33,34,35,36]. Unfortunately, the primary line of treatment for OSA, i.e., continuous positive airway pressure (CPAP), has resulted in being disappointingly ineffective in protecting against OSA-induced CVD [37,38,39].

RAAS activation is one of the best-studied hormonal systems involved in the pathophysiology of systemic hypertension and of cardiovascular disease [40,41,42]. Briefly, active renin acts upon angiotensinogen to generate angiotensin I that is cleaved by the angiotensin-converting enzyme to the physiologically active angiotensin II (Ang II). As a main effector of the RAAS system, Ang II exerts many of its effects via type I angiotensin receptors (AT1R) [43]. Another important component of the RAAS, aldosterone, exerts crucial endocrine functions such as regulating fluid volume, sodium, and potassium homeostasis, and primarily acts on mineralocorticoid receptors (MR) in the renal distal tubules [43]. Discovery of an important MR-dependent signaling in the cardiovascular system has resulted in major expansion of the scope of the research on aldosterone-mediated cardiovascular effects [44,45,46]. Indeed, MR activation mediates genomic and non-genomic effects via second messenger systems, and via crosstalk with multiple receptors that underlie downstream vascular biological and (patho)physiological functions, particularly vascular inflammation, oxidative stress, fibrosis, and cardiac remodeling and hypertrophy [47,48]. Clinical and experimental trials have shown that targeted blockade of MR receptors can reduce blood pressure, alleviate oxidative stress and inflammation, improve vascular function and insulin sensitivity, and protect against atherosclerosis, HF, MI, and AF [49,50,51,52,53,54,55,56,57,58]. In other words, MR blockers alleviate a similar spectrum of conditions that are associated with OSA.

Although the relationship between OSA and RAAS activation is not fully understood [59], a bidirectional association exists between the two conditions, whereby OSA can trigger RAAS activation, and the latter can then worsen the prognosis and morbidities associated with OSA [59]. Nevertheless, both conditions share similar mechanisms and processes that underlie cardiovascular disturbances and increase overall mortality, as well as cardiac-specific mortality [5,7,26,44,45]. Despite this rather conspicuous relationship, very few studies have evaluated the effects of MR antagonism in OSA, and of these few studies, most have focused on resistant hypertension in OSA patients [60,61,62]. Of note, these studies have been conducted in light of the disappointing outcomes reported by the few large, randomized control trials (RCTs) that showed minimal to no benefit of CPAP treatment in protecting against OSA-mediated CVD [37,38,39]. Moreover, there are no experimental studies using available OSA models evaluating the role of MR signaling in cardiac and vascular tissues, and the potential benefit of MR antagonism on cardiovascular outcomes. This review focuses on the relationship between the OSA and MR activation, mechanisms shared by both conditions in promoting CVD, and the potential cardiovascular benefits of MR antagonism, thus highlighting MR as a potential therapeutic target for OSA-induced CVD.

## 2. OSA and CVD

As mentioned above, OSA is a highly prevalent and a serious chronic sleep-related breathing disorder affecting up to one billion people globally [1]. The chronic physiological stresses from recurrent airway obstruction can directly induce cardiovascular morbidity and mortality [5,6,7,26]. Indeed, it is estimated that more than 40% of patients with CVD have OSA [1]. However, OSA remains critically underdiagnosed and undertreated, which raises the risk of developing OSA-related long-term cardiovascular consequences [63]. Several factors hinder adequate OSA screening and diagnosis, including confounding variables and logistical and financial barriers. The gold standard test, polysomnography, can be costly with long waiting times [64,65,66]. Several questionnaires were developed to screen for OSA based on risk factors (male sex, older age, obesity, craniofacial abnormalities, and genetics) [67]. However, these OSA screening questionnaires have limited specificity and poor diagnostic accuracy, especially in patients with underlying CVD [68,69].

OSA has been associated with multiple cardiovascular complications including hypertension, HF, CAD, arrhythmias, and CV mortality [1,5,6,63]. OSA and hypertension often coexist with an estimated 50% of OSA patients having hypertension and 30% of hypertensive patients also having OSA [70,71,72]. In patients with untreated OSA, the risk of developing hypertension is increased 2–3-fold [73], while around 65–80% of patients with resistant hypertension may suffer from OSA [74,75]. The largest impact of OSA on blood pressure is due to dampening of the normal nocturnal fall in blood pressure (dipping) [76]. In HF, The Sleep-Disordered Breathing in Heart Failure (SchlaHF) Registry reported the prevalence of moderate to severe OSA to be around 50% in men and 36% in women with stable, symptomatic heart failure with reduced ejection fraction (HfrEF) [77]. It is also estimated that OSA is present in more than 50% patients with HfpEF [78]. OSA can increase the risk of symptom progression and mortality in HF patients with increased risk of death by 16.1% for each hypoxemic hour (time with oxygen saturation less than 90%) [36,79]. AF has also been associated with OSA with a ranging prevalence between 21% and 74% [35,80]. OSA can independently induce left atrial structural changes and enlargement that predisposes OSA patients to AF [81]. The severity of OSA in directly proportional to increased risk of AF and the presence of OSA has been also shown to inhibit the efficacy of antiarrhythmic treatments in AF patients [82,83]. OSA have been also associated with bradyarrhythmias, supraventricular and ventricular tachyarrhythmias, and sudden cardiac arrest [34,84,85]. It is estimated that 38% to 65% of patients with CAD have OSA and that OSA is present in nearly 50% of patients requiring percutaneous coronary intervention [3,4]. The risk of developing CAD increases substantially in the presence of OSA. In fact, OSA increases the risk for development of MI, coronary revascularization events, or CV death by 2-fold, independent of other risk factors [86]. OSA has been reported to independently enhance the risk of subclinical CAD measured by coronary calcification score and can independently predict the risk of atherosclerotic plaque progression and vulnerability [32,33]. In the SHHS (Sleep Heart Health Study), the all-cause mortality was significantly higher among male patients with severe OSA [87]. Similarly, The MESA (Multiethnic Study of Atherosclerosis) showed that OSA was associated with 2.2 higher incidence of cardiovascular events and a 2.4 increase in mortality over a 7.5-year follow-up period [88].

Repeated nocturnal hypoxic episodes in OSA can initiate a plethora of pathophysiological mechanisms that promote cardiac and vascular disease [3,4,5,6,7,24,26]. OSA-induced IH and hypercapnia elicit chemoreflex-mediated enhancements of sympathetic activity and RAAS activation leading to vasoconstriction and subsequent increases in blood pressure surges and hemodynamic disturbances [19]. IH and sleep fragmentation (SF) are also considered major activators of pro-oxidant and pro-inflammatory pathways leading to systemic inflammation and dysfunction, particularly in the cardiovascular system. All the aforementioned mechanisms contribute to endothelial dysfunction and an enhanced prothrombotic state, initiating the atherosclerotic process and cardiac remodeling [5,7,31]. Additionally, resultant intrathoracic swings can influence cardiac transmural gradients and affect ventricular function, leading to increased wall stress and impaired diastolic function [89]. Evidence from clinical and experimental models of OSA, especially those involving IH exposures, show that lipid, protein, and DNA oxidative stress markers are elevated with activated NADPH oxidases, xanthine oxidase, and mitochondria being major sources [16,17,18,26,90,91,92,93,94,95]. Other studies have also shown enhanced NF-ƙB activation, aggravated proinflammatory cytokine and adhesion molecule production, and profibrotic signaling [13,14,15,16,90,96,97,98,99,100,101,102,103,104,105,106]. Moreover, OSA and experimental IH have shown to disrupt lipid and glucose homeostasis, leading to insulin resistance and metabolic disturbances [11,12,107,108,109,110,111,112]. Collectively, the overwhelming evidence implicates OSA and its preclinical models in the processes favoring the occurrence of atherosclerosis and CVD.

## 3. OSA Management with CPAP

CPAP is the mainstay treatment for OSA due to its positive impact on symptoms and quality of life, which provides a constant level of positive pressure across inspiration and expiration [113]. Although CPAP treatment has been shown to reduce AHI and mitigate or even abolish the episodic hypoxemia, both of which are predictors of CVD, it failed to reduce the rates of cardiovascular complications in OSA patients [89]. The SAVE (Sleep Apnea Cardiovascular Endpoint) study randomized over 2700 OSA patients with a history of CVD to usual care plus CPAP versus usual care alone and failed to demonstrate conclusive evidence of significant reductions in the primary endpoint (composite CVD) after 3.7 years of follow-up with ongoing CPAP treatment [39]. Similar negative findings were found in The ISAACC (CPAP in Patients With Acute Coronary Syndrome and OSA) and The RICCADSA (Randomized Intervention with CPAP in CAD and OSA) trials, which investigated the effectiveness of CPAP on the putative reduction in CV complications following acute coronary syndrome and PCI, respectively [37,38]. A recent meta-analysis of nine randomized trials showed that CPAP treatment did not prolong survival or reduce cardiovascular events in adults with OSA and CVD [114]. Poor CPAP adherence (minimal adherence being >4 h per night) was identified as the main likely contributor to the negative outcomes of the trials [37,38,114,115]. Moreover, incidents of cardiovascular complications, such as CAD, are substantially increased by OSA, especially in patients without a previous history of CAD [37]. This suggests that advanced atherosclerotic vascular pathological processes are more likely to be irreversible with OSA treatment, a finding that was recapitulated in experimental IH delivered to mice for extended periods of time [116]. Ultimately, the need for additional adjuvant therapies aimed at CVD induced by OSA are needed. In the next sections, we discuss the impact of MR activation in CVD and the potential usefulness of MR antagonists in targeting OSA-induced CVD.

## 4. MR Structure and Function

MR (NR3C2) is a member of a steroid-activator transcription factor superfamily that preserves structural similarities to the glucocorticoid receptor (NR3C1, GR) and the progesterone receptor [117]. Similar to other nuclear receptors, MR contains DNA- and ligand-binding domains, in addition to an amino terminal domain and hinge region [48]. Under basal conditions, MR is predominantly present in the cytosolic fraction as part of a heterocomplex with multiple chaperones that are important for facilitating ligand binding and cytoplasmic–nuclear translocation [118,119]. To allow DNA binding, MR dissociates from its chaperones and forms homodimers or heterodimers with GR in certain conditions that can then result in different transcriptional responses [120,121]. Once the MR DNA-binding domain binds to hormone response elements, gene transcription ensues. MR can also form complexes with other transcription factors rather than directly bind to DNA itself [122].

Aldosterone is the major mineralocorticoid hormone synthesized in the zona glomerulosa of the adrenal gland under the influence of the RAAS, adrenocorticotropic hormone (ACTH), and extracellular potassium levels [123]. The main function of aldosterone is to regulate the fluid and salt balance via sodium transport machinery in renal epithelial tubular cells [124]. Besides aldosterone, MR has a strong affinity for glucocorticoids that are significantly more abundant in the circulation [125]. However, aldosterone can remain bound to MR for more prolonged periods than cortisol, which stabilizes MR conformation. As a mechanism to confer the specificity of MR to aldosterone, the enzyme 11-beta-hydroxysteroid dehydrogenase type 2 (HSD11B2) is co-expressed with MR and can metabolize cortisol to cortisone, which is unable to bind to or activate MR [126]. Indeed, HSD11B2 deficiency or inhibition can result in hypertension and hypokalemia due to activation of cortisol by renal MR [127]. However, under cellular oxidative stress and inflammation, glucocorticoids can activate MR [128,129]. Thus, distribution of HSD11B2 expression, ligand availability, and cellular redox status can influence the response of the MR to ligand activation.

MR utilizes several mechanisms to promote cellular changes and regulates a multitude of genes, with the majority of such genes being involved in electrolyte homeostasis in renal epithelial tissues [47]. Other MR target genes can be expressed upon stimulation by aldosterone in other tissues, including vascular and cardiac tissues that can affect signal transduction, redox status, cellular structure, adhesion, and migration [130,131]. Due to the slow nature of gene transcription, MR can also trigger faster non-genomic mechanisms, for instance in response to acute changes in fluid homeostasis [47]. MR rapid signaling which occurs through second messenger systems includes: (i) Mitogen-associated protein kinases (MAPK) for cell proliferation or apoptosis and electrolyte handling [132], (ii) Phosphatidylinositide 3-kinases (PI3K) involved in electrolyte handling and vasomotor function [133], (iii) Protein kinase C (PKC) and protein kinase D for renal epithelial cells and cardiomyocyte electrolyte handling [134,135]. Other rapid non-genomic MR-dependent aldosterone effects can be initiated at the plasma membrane where the classical MR is associated with scaffolding proteins to the cytosolic side of the plasma [136]. Striatin and caveolin-1 have been recently identified as candidates for such scaffolding proteins [137,138]. As a result, transactivation of many receptor tyrosine kinases and G-protein-coupled receptors that are located close to the MR scaffolding proteins have been reported [136]. Additionally, MR can initiate crosstalk with other cytosolic signaling pathways affecting genomic signaling, such as nuclear factor of activated T-cells (NFAT) and cAMP-response element-binding protein (CREB) [139,140]. Lastly, genomic MR signaling can be influenced by epigenetic mechanisms and by posttranscriptional regulation by microRNAs [141,142] (Figure 1).

## 5. Pathological Mechanisms of MR Activation

*Oxidative stress*: MR activation can lead to reactive oxygen species (ROS) generation, particularly through the upregulation and activation of NADPH oxidases (NOX) [47]. NOX is a family of membrane-bound enzymes that produces superoxide anions and is abundantly present in endothelial cells, vascular smooth muscle cells (VSMCs), cardiomyocytes, and leukocytes [143,144]. The enhanced ROS production mediated by MR activation can lead to excessive activation of proinflammatory and profibrotic signaling through activator protein 1 (AP-1) and nuclear factor kappa B (NF-ƙB) [145,146]. Aldosterone can rapidly and persistently activate NOX within minutes, suggesting a non-genomic regulation of NOX [147]. Indeed, MR activates NOX through c-Src and downstream Ras-related C3 botulinum toxin substrate (Rac1) activation [148,149]. MR can also increase NOX-dependent ROS production through crosstalk with epidermal growth factor receptor (EGFR) and AT1R [149]. Additionally, MR can upregulate NOX by enhancing synthesis of NOX cytosolic subunits in endothelial cells and cardiomyocytes [150,151]. In animal models of hypertension using aldosterone and salt, the levels of oxidized lipids were elevated in urine, plasma, kidney, and hearts [152,153,154], with levels of 8-isoprostane levels reaching 10-fold higher in hearts of aldosterone-treated animals when compared to controls [153]. Increased superoxide anion production has been also reported in multiple tissues in aldosterone-treated animals [155,156]. Furthermore, the expression of NOX was elevated in hearts and vasculature of aldosterone-treated rats [151,157]. Treatment with the MR antagonist spironolactone lowered free radical production, confirming the involvement of MR in aldosterone-induced oxidative stress [158,159,160,161,162]. In patients with essential hypertension, who often exhibit hyperaldosteronism, the levels of oxidative stress markers were elevated [163,164]. Treatment with spironolactone lowered urinary oxidative stress markers in patients with chronic kidney disease and reduced superoxide anion production in isolated macrophages from patients with congestive HF [165,166].

*Inflammation:* Persistent MR activation is associated with enhanced inflammatory responses that can eventually lead to tissue remodeling and fibrosis [167,168,169,170]. MR regulates transcription of genes involved in the recruitment of adhesion molecules via platelet-derived growth factor receptor (PDGFR) and c-Src activation [171]. It can also increase production of inflammatory cytokines, including tumor necrosis factor alpha (TNF-α) and interleukin 6 (IL-6), via regulation of NF-ƙB transcription, which is further enhanced by serine/threonine-protein kinase 1 (SGK1) [172,173]. Constant exposure to aldosterone for 14 days is associated with perivascular and cardiac inflammation, as reflected by leukocyte recruitment, adhesion, and infiltration [174,175]. MR activation induced intracellular adhesion molecule 1 (ICAM1) and P-selectin expression in endothelial cells and in VSMCs and induced vascular endothelial growth factor (VEGF) recruitment of monocytes [176,177,178]. AT1R has been shown to be required for expression of inflammatory genes in the presence of aldosterone in VSMCs since Agtr1a-null mice were protected against aldosterone-induced vascular dysfunction [179,180]. Aldosterone can also induce C-reactive protein (CRP) expression in VSMCs in vitro and in vivo that is prevented by MR antagonism [181]. One of the other mechanisms by which aldosterone induces inflammation is through activation of the NLR family pyrin domain containing 3 (NLRP3) inflammasome that can lead to caspase-1-dependent induction of the pro-inflammatory cytokines interleukin 1 beta (IL-1β) and interleukin 18 (IL-18) [182]. Rats infused with aldosterone exhibited increased macrophage infiltration of the kidneys and increased expression of inflammasome markers that were attenuated with immunosuppressant treatment [183]. NLRP3 deletion prevented aldosterone-mediated expression of adhesion molecules and adherence of macrophages to aortic segments in mice [182]. A recent study showed that TNF-α and IL-6 were elevated in patients with primary aldosteronism (PA) when compared to normotensive individuals or to patients with essential hypertension [184]. Treatment of hypertensive patients with eplerenone, an MR antagonist, reduced many proinflammatory mediators, including monocyte chemoattractant protein 1 (MCP-1) and interleukin 8 (IL-8) [185].

*Sympathetic activation:* MR can be expressed extra-renally in tissues such as the heart, brain, and peripheral sensory neurons [186,187,188]. Despite the well-established role of MR activation in RAAS-induced sympathetic activation, aldosterone has shown to affect the parasympathetic nervous system through inhibition of baroreceptor discharge and bradycardic responses to pressor stimuli [189,190]. It can also activate the sympathetic system and augment its activity by attenuating catecholamine uptake in myocardial tissues [191,192]. In the hypothalamic paraventricular nucleus (PVN) region, MR can contribute to Ang II-induced hypertension [193]. Indeed, central infusion of MR antagonists can prevent neuronal activation in the PVN by Ang II and the associated increases in blood pressure [194]. Moreover, local expression of MR, 11β-HSD2, and aldosterone in the neurons of the cardiac autonomic nervous system has been recently discovered, suggesting a putative modulatory role of MR on cardiac endogenous neuronal activity [195]. Overall, MR may contribute directly to sympathetic activation or parasympathetic inhibition, leading to increased blood pressure and cardiovascular dysfunction.

*Endothelial dysfunction*: Endothelial dysfunction is the main pathological process preceding atherosclerosis and is manifest as impaired vasodilation and/or enhanced vasoconstriction [196]. In vitro, aldosterone reduces nitric oxide (NO) production in endothelial cells via inhibition of endothelial nitric oxide synthase (eNOS) activity. Aldosterone-induced eNOS inhibition is mediated through increased RhoA kinase activity and consequent protein kinase B (Akt) signaling inhibition, or via increased protein phosphatase 2A (PP2A) activity, leading to eNOS dephosphorylation [157,197]. Vascular endothelial MR activation increases oxidative stress [198]. Aldosterone-induced ROS overproduction can reduce NO bioavailability and lead to the production of the pernicious reactive nitrogen specie peroxynitrite that can oxidize lipids, proteins, and DNA (Figure 2) [199,200]. It can also lead to tetrahydrobiopterin (BH_4_) depletion, an important cofactor for NO generation by eNOS, resulting in more production of ROS instead of NO [157,201]. Moreover, aortic expression of cyclooxygenase 2 (COX-2) is elevated in aldosterone-treated rats [202,203]. COX-2 can generate vasoactive prostanoids and ROS that not only impair vasodilation but can also enhance vasoconstrictive responses (Figure 2) [203,204]. Aldosterone can also indirectly impair endothelium-dependent vasodilation through glucose-6-phosphate dehydrogenase (G6PD) reduction, exacerbating oxidative stress and impairing NO production [205]. MR activation can also exert NO-independent effects on vasomotor responses. Aldosterone can increase endothelial cell volume and tension though NHE-1 and/or epithelial sodium channel (EnaC) activation, with the latter also affecting eNOS activity and NO production [206,207,208]. However, endothelial response to MR is vastly heterogeneous depending on sex, vascular bed, steroid use, duration of exposure, and environmental context [209,210,211]. For instance, MR deletion in the endothelium had no effect on mesenteric vasomotor responses but significantly mitigated vasoconstrictor responses to endothelin-1 in coronary arteries [210]. Moreover, endothelial MR deletion in females was more protective than in males in the context of diet-induced obesity [212]. The heterogeneity of MR responses extends to signaling through other receptors such as AT1R signaling that is important for aldosterone-mediated endothelial dysfunction. Indeed, inhibiting or knocking out of AT1R in mice blunted endothelial dysfunction after treatment with aldosterone [179]. In humans, there is a strong association between aldosterone levels and impaired flow-mediated dilatation (FMD), a gold-standard indicator of endothelial function [213]. FMD is also inversely correlated with plasma aldosterone and urinary aldosterone secretion in patients with resistant hypertension [214]. In PA patients, FMD is inversely correlated with PA phenotype severity and lower numbers of circulating endothelial progenitor cells that also exhibit reduced migratory potential [215,216,217]. Despite the controversial benefit of MR antagonist on FMD, a recent meta-analysis showed that FMD increased in 11 trials including 570 patients after treatment with MR antagonists (mean difference, 1.18% [95% CI, 0.14 to 2.23], *p* = 0.03) [218].

*Fibrosis:* MR activation can regulate multiple pro-fibrotic and pro-hypertrophic genes via second messenger systems. For instance, MR can induce cardiac fibroblast proliferation and hypertrophy-associated genes as myosin heavy chain beta (MHC-β) through extracellular signal-regulated kinase (ERK) signaling [219]. Additionally, MR activation can promote cardiomyocyte production of connective tissue growth factor (CTGF) via p38 MAPK signaling [220]. AT1R transactivation by MR is also involved in the production of fibrotic and hypertrophic genes including transforming growth factor beta (TGF-β) and alpha-smooth muscle actin (α-SMC) through ERK and c-Jun N-terminal kinase (JNK) signaling [221]. Moreover, cardiac MR transactivation of EGFR can increase sodium–hydrogen antiporter 1 (NHE-1) activity that results in sodium accumulation promoting calcium influx and subsequent facilitation of generation of pro-hypertrophic and pro-fibrotic factors [222]. Genomically, increased SGK1 by MR activation upregulates cardiac CTGF [223]. Hyperaldosteronism in mice exacerbated hypertension-induced fibrosis through activation of inflammation/galectin-3-induced fibrosis and inhibition of antifibrotic factors such as B-type natriuretic peptide (BNP) [224]. Cardiomyocyte-specific MR-null mice after MI showed improved reparative scar formation and reduced infarct size and reactive fibrosis in the viable ventricle wall [225].

*Arterial stiffness:* Arterial stiffness reflects the attenuation of arterial distensibility due to the combination of smooth muscle cells contraction (active stiffness) and changes of inert components of the vascular wall (passive stiffness) [226]. Arterial stiffness can be assessed by pulse wave velocity (PWV) and by the augmentation index [226]. In animal studies, aldosterone augmented active stiffness by enhancing vascular myogenic tone and vasoconstrictive agent-dependent contraction that was reduced in VSMC knockout mice [227]. Aldosterone can also induce passive stiffness through modulation of the expression of several vascular genes involved in extracellular matrix remodeling including CTGF and metalloproteinases [130]. Furthermore, aldosterone increases collagen deposition in the arterial wall of multiple animal models via several pathways including galectin-3 [228], neutrophil gelatinase-associated lipocalin (NGAL) [229], ENAC [207], and NOX [230] (Figure 2). It has been shown that aldosterone can induce osteogenic phenotype in VSMCs through activation of alkaline phosphatase and NOX, reducing VSMCs autophagy, and induction of inflammatory pathways, thereby increasing calcification of the vascular wall [231,232,233,234]. Spironolactone treatment mitigated the progression of calcification in uremic rats [235]. High aldosterone has been associated with increased PWV in patients newly diagnosed with hypertension, independent of blood pressure [236]. Treatment with spironolactone significantly reduced both PWV and the augmentation index in patients with hypertension [237]. In PA patients, a meta-analysis showed that PWV is higher when compared to matched patients with essential hypertension, but no differences emerged in the augmentation index [238]. Adrenalectomy significantly reduced PWV and augmentation index in patients with PA [239]. A recent meta-analysis showed that MR antagonists reduced PWV in 11 trials including 515 patients when compared to controls (mean difference, −0.75 m/s [95% CI, −1.12 to −0.39], *p* < 0.00001) independent of blood pressure reduction [218]. MR antagonist treatment also reduced the augmentation index compared with controls in 5 trials including 283 patients (mean difference, −6.74% [95% CI, −10.26 to −3.21], *p* = 0.0002) [218].

*Insulin resistance:* Insulin resistance is characterized by the reduced ability of insulin to activate insulin signaling and glucose uptake [240]. Although the main targets of insulin are skeletal muscle, liver, and adipose tissues, insulin receptors are also present in vascular cells, among many other cells [240]. Diminished insulin signaling in vascular tissues leads to reduced glucose uptake and insulin resistance [241]. Animal studies have shown that excessive MR activation impairs insulin signaling and induces vascular stiffness, mainly by aggravating oxidative stress and inflammation [242,243,244,245,246]. In the endothelium, aldosterone can enhance the upregulation and translocation of EnaC to the cell wall, which induces vascular stiffness, impairs insulin-mediated capillary recruitment, and enhances insulin resistance (Figure 2) [245]. MR activation can also activate the mammalian target of the rapamycin/ribosomal protein S6 kinase beta-1 (mTOR/S6K1) signaling pathway, which induces vascular insulin resistance via increased serine phosphorylation insulin receptor substrate 1 (IRS1), leading to reduced translocation of glucose transporter 4 (GLUT4) to the cell membrane and glucose uptake [247]. Thus, reduced insulin signaling can lead to impaired PI3K/Akt signaling and consequent inhibition of eNOS phosphorylation and NO production, and the hyperinsulinemia associated with such events can increase the release of vasoconstrictive substances, ultimately leading to impaired vasodilation and enhanced vasoconstriction (Figure 2) [246,247]. MR blockade with spironolactone or with mice harboring endothelial-specific MR knockout abolished diet-induced aortic and mesenteric arterial stiffness and microvascular dysfunction, improved insulin-mediated eNOS activity, and suppressed endothelial EnaC activation [210,242,245]. Furthermore, spironolactone decreased the expression of multiple inflammatory cytokines and decreased oxidative stress markers with accompanying increases in vascular insulin sensitivity in female mice fed a Western diet [242]. Despite the promising results from preclinical data on spironolactone and eplerenone on glucose metabolism insulin sensitivity, clinical studies have shown no benefit from MR antagonism on glucose metabolism in relevant patient cohorts. Data from the Candesartan in Heart Failure Assessment of Reduction in Mortality and Morbidity (CHARM) clinical study showed that MR antagonists may be involved in increased risk for the development of diabetes [248]. Additionally, a recent meta-analysis of 18 trials assessing spironolactone effects on fasting glucose and insulin, hemoglobin A1c, and homeostatic model assessment (HOMA)-insulin resistance (IR) showed that spironolactone increased HbA1c but had no clear effect on fasting glucose, HOMA-IR, and insulin levels [249]. However, newer non-steroidal and more selective MR antagonists may prove to have a favorable impact on glucose and insulin [250].

*Cardiac electrical remodeling*: MR activity can influence cardiomyocyte electrolyte handling, the action potential, and cardiac contractility [47]. Low-voltage-activated T-type channels that regulate rapid activation and slow deactivation, and L-type dihydropyridine channels that activate slower but more rapidly deactivate than T-type channels, are both crucial for pacemaker activity and action potential propagation [251]. MR activation increases calcium current through both of these types of calcium channels [251,252]. Indeed, cardiomyocyte calcium status is linked to transmembrane sodium concentrations [253]. Aldosterone can raise sodium influx rapidly via multiple channels via transactivation of many second messenger systems, such as increased EGFR-dependent NHE-1 activation [222,254]. In addition to increasing sodium influx, NHE-1 can regulate intracellular pH and cell volume, leading to cardiomyocyte alkalinization increasing myofilaments response to calcium [255]. At the structural level, enhanced calcium load can induce profibrotic pathways in atrial tissues, leading to atrial remodeling and dilatation and consequent atrial fibrillation [256]. Cardiac overexpression of MR in mice increases action potential duration and ventricular arrhythmia due to aberrant release of calcium from the sarcoplasmic reticulum [257]. MR knockdown with interfering RNA blocked slow force response accompanied by reduced NHE-1 activity [258].

## 6. Cardiovascular Consequences of MR Activation

*Coronary artery disease*: CAD is the consequence of the atherosclerotic processes that are driven by activated endothelium and macrophage recruitment followed by progressive engulfment of macrophages with oxidized LDL, forming foam cells. Thus, vascular oxidative stress and inflammation are very crucial pathological stimuli for plaque formation, the major cause for CAD (Figure 3) [259]. Aldosterone can activate endothelial cells and stimulate VSMCs, increasing leukocyte and monocyte adhesion and infiltration [176,178,260]. Once in the intima media layer, monocytes differentiate into M1 macrophages characterized by a proinflammatory action that is facilitated by aldosterone [175,182]. In Apolipoprotein E (ApoE) knockout mice, aldosterone infusion increased the infiltration of macrophages in the atherosclerotic plaque, increased plaque size and lipid accumulation in the plaque [176,261,262]. MR antagonism reduced plaque size, lipid peroxides, and oxidation of LDL within the plaque [263,264]. Moreover, specific MR deletion in VSMC improved left ventricular function after MI via preservation of coronary flow reserve [265]. In humans, studies have shown a clear association between aldosterone levels and atherosclerosis in the coronary arteries [266]. Elevated aldosterone levels are independently associated with an increased risk of acute cardiac ischemic events, higher rates of cardiovascular events, cardiovascular mortality, and overall mortality in patients with acute MI [267,268]. Patients with PA exhibit more pronounced vascular inflammatory phenotype when compared to patients with essential hypertension [269,270]. Results from a meta-analysis showed that patients with PA are at a 1.77-fold higher risk of developing coronary artery disease compared to patients with essential hypertension [271]. Adrenalectomy and treatment with MR antagonists reduced the risk of cardiovascular disease in PA patients [272]. In large, randomized control trials, the REMINDER trial (Impact of Eplerenone on Cardiovascular Outcomes in Patients Post MI) showed that eplerenone reduced the primary composite outcome (cardiovascular events and mortality, reduced left ventricular function, and prolonged hospitalization) [273]. In the ALBATROSS trial, (Aldosterone Lethal Effects Blocked in Acute MI Treated With or Without Reperfusion to Improve Outcome and Survival at Six Months Follow-Up), intravenous canrenone administration followed by oral spironolactone for 6 months did not improve the composite primary outcome, but exerted a benefit in mortality among patients with MI with ST-segment elevation [274]. In the DIDELIO-DKD trial (Finerenone in Reducing Kidney Failure and Disease Progression in Diabetic Kidney Disease), finerenone reduced the risk of the composite cardiovascular outcome compared with placebo in patients with or without cardiovascular disease [275].

*Heart Failure*: Hemodynamic overload and neurohumoral mechanisms can lead to cardiac hypertrophy and fibrosis. In the presence of MI, the loss of cardiomyocytes exceeding the cardiac regenerative capacity can lead to fibrotic tissue formation and consequent ventricular remodeling [276]. Excessive MR stimulation plays an integral role in both situations, leading to left ventricular systolic and diastolic dysfunction, heart failure, and increased overall cardiovascular mortality (Figure 3) [277]. In murine models, aldosterone infusion increases perivascular and interstitial fibrosis through oxidative stress and inflammatory pathways and alteration of extracellular matrix deposition through metalloproteinases inhibition and increased collagen deposition [278,279,280,281,282,283]. Direct effects of aldosterone on cardiac hypertrophy are mediated via increased blood pressure and hemodynamic overload [284]. Aldosterone infusion increases myocardial hypertrophy through cardiotropin-1, plasminogen activator inhibitor 1 (PAI-1), and circadian clock proteins [285,286,287]. Most of these mechanisms can lead to cardiac fibrosis and contribute further to ventricular hypertrophy and heart failure [285,287,288]. Patients with PA exhibit enhanced signs of cardiac fibrosis and increased rates of left ventricular mass and hypertrophy [271,289,290]. There is a correlation between the severity of left ventricular hypertrophy and the level of autonomous aldosterone secretion [291]. Systolic and diastolic function are also compromised in PA patients who display increased risk of heart failure than patients with essential hypertension [271,292,293]. MR antagonist treatment and adrenalectomy partially reverse left ventricular hypertrophy and restore diastolic function [272]. MR antagonists have been evaluated in patients with heart failure with preserved ejection fraction (HfpEF) and HfrEF [294]. In patients with HfrEF, the EPHESUS trial (Eplerenone Post-Acute MI Heart Failure Efficacy and Survival Study) and the RALES trial (Randomized Aldactone Evaluation Study) have demonstrated that the addition of up to 50 mg of eplerenone and 25 mg of spironolactone to the standard therapy significantly reduces cardiac and overall mortality [295,296]. The ARTS-HF phase 2b trial (Mineralocorticoid Receptor antagonist Tolerability Study-HF) compared the efficacy of finerenone vs. eplerenone in patients with HfrEF and diabetes or chronic kidney disease, showing similar reduction effects on natriuretic peptide with a greater safety profile [297]. Spironolactone improved diastolic function in patients with HfpEF and diastolic dysfunction [298,299]. In the TOPCAT trial (Treatment of Preserved Cardiac Function Heart Failure With an Aldosterone Antagonist), spironolactone did not improve the primary composite outcome of cardiovascular mortality [300]. However, a reduced primary outcome composite after a *post hoc* analysis was present only in patients with resistant hypertension [300]. A recent individual-patient-data meta-analysis of three large RCTs showed that spironolactone improved cardiac structure and function of patients with HfpEF [301].

*Atrial fibrillation*: Like all tachyarrhythmias, the onset and maintenance of AF is dependent on three key mechanisms: automaticity, triggered activity, and re-entry. Automaticity refers to when an excitable tissue spontaneously depolarizes, triggered activity refers to when additional impulses triggered by afterdepolarizations relate to calcium overload, and re-entry refers to when waves of depolarization circle around an obstacle reinitiating continuously [256]. Experimental studies show that aldosterone has the ability to alter the electrophysiological properties of cardiomyocytes, enhancing cytosolic calcium load and promoting cardiac arrythmias (Figure 3) [302,303]. Chronic aldosterone infusion in rats enhances P-wave duration, activation time of the right atrium, and atrial anisotropy of conduction, facilitating re-entry mechanisms and AF stabilization [304]. Furthermore, remodeling of atrial tissue through increasing fibroblast proliferation and collagen deposition in the atrium contributed further to AF maintenance, increasing the time of spontaneous conversion [304,305]. MR antagonism in a tachy-paced sheep model partially reduced atrial fibrosis and dilatation with subsequent reduction in AF progression [306]. In patients with PA, the risk of AF is 3.52-fold higher than patients with essential hypertension [271]. It is estimated that 42% of AF patients have PA which warrants the screening of these patients for PA [307,308]. Although adrenalectomy reduces AF incident risks in PA patients, the use of MR antagonists failed to reduce the risk of AF [309]. The EMPHASIS-HF trial has shown that eplerenone significantly reduced the new-onset AF in patients with HfrEF [310]. The RACE-3 trial (Routine Versus Aggressive Upstream Rhythm Control for Prevention of Early Atrial Fibrillation in Heart Failure) showed that in the intervention group (MR antagonists, statins, angiotensin converting enzyme inhibitors and/or receptor blockers, and cardiac rehabilitation) significantly improved sinus rhythm maintenance in HF patients with AF when compared to conventional therapy. Interestingly, 85% of patients undergoing interventional therapy used MR antagonists versus only 4% in the conventional therapy group [311]. As a result, the European Society of Cardiology Guidelines of 2020 have introduced MR antagonists as a potential non-antiarrhythmic medication for AF therapy [312].

## 7. Is MR a Mediator of OSA-Induced CVD?

As mentioned earlier, MR activation-induced CVD can be induced through multiple mechanisms similar to those influenced by OSA, while OSA can directly activate RAAS and elevate renin, Ang II, and aldosterone levels [5,7,26,47,313]. Thus, it is plausible that OSA can mediate not only resistant hypertension and PA, but also atherosclerosis and CVD through MR activation, at least partially (Figure 4). The Endocrine Society has noted OSA in the presence of hypertension as one of the groups with high prevalence of PA, and now recommends screening of PA among this group of patients [314]. Experimental IH and SF have been shown to increase plasma levels of aldosterone and renin and stimulated release of ACTH from the pituitary and enhance cortisol secretion [315,316,317]. Both ACTH and RAAS act synergistically to regulate aldosterone pulse wave [318,319]. While elevated cortisol concentrations control the pulse amplitude during the daytime, RAAS plays a substantial role at night when plasma cortisol concentration is low [318]. Additionally, it is well established that obesity is a major risk factor for OSA and that 70% of OSA patients are obese [320]. The dysfunctional adipose tissue in obesity is considered an important source of RAAS hormone secretion that is independent from the classical RAAS activation [62,321]. It has been demonstrated that adipocytes can release adipokines responsible for stimulation for aldosterone secretion from adrenocortical cells [316,322]. Overall, evidence points to OSA-mediated RAAS activation. However, it is also suggested that excess aldosterone can worsen the clinical course of OSA potentially through increased salt and water retention, leading to rostral fluid shifts and para-pharyngeal edema and exacerbated upper airway obstruction and collapsibility, further worsening the severity of OSA [323,324,325]. Aldosterone can also act centrally to increase brain RAAS activity and oxidative stress, which may cause abnormal regulation of central breathing mechanisms, worsening OSA prognosis [326]. Thus, early OSA diagnosis and management and MR inhibition are crucial for preventing or ameliorating OSA-induced CVD.

Several studies have evaluated the use of MR receptor antagonists on resistant hypertension in OSA. Two RCTs reported positive modulation in OSA severity and lowering blood pressure using spironolactone, but due to the small effect size and risk of bias, no definitive conclusions can be drawn from the available data [60,61]. Other studies used eplerenone in OSA and found that the treatment reduced blood pressure, arterial stiffness, and left ventricular hypertrophy in OSA patients [327,328]. These promising results showcase the significance of MR antagonists, not only as an antihypertensive medication, but also as a treatment for OSA-induced CVD. Conversely, several studies with small sample size and short treatment duration with CPAP did not reveal the anticipated reduction in the RAAS components [319,329,330,331,332,333], further justifying the need to include MR antagonists as an add-on adjuvant therapy in OSA patients. However, there is a lack of experimental data and of large RCTs that have assessed the benefits and safety of MR antagonists as part of long-term treatment of OSA-induced CVD. As described earlier, MR activation shares multiple pathophysiological mechanisms with OSA, all of which can be targeted by MR antagonism [7,313]. Extensive experimental IH and SF studies are needed, evaluating multiple in vivo and in vitro cardiovascular parameters while using different MR antagonists as well as specific transgenic animals with cell-specific targeted disruption of MR, to better understand the mutually dependent effects and interactions between RAAS and OSA and further explore the viability of MR-targeted treatment on OSA cardiovascular outcomes.

## 8. Conclusions

OSA is a chronic and highly prevalent condition that is associated with vascular dysfunction, cardiac remodeling, and overall cardiovascular risk and mortality. MR is essential for fluid homeostasis and balance. However, excessive MR activation in cardiovascular tissues may promote atherosclerosis and cardiac dysfunction through similar mechanisms induced by OSA. OSA can trigger RAAS activation and subsequent excessive MR activation. Thus, enhanced MR activation in OSA patients could contribute to OSA-induced CVD. However, MR influences on CVD in OSA has been insufficiently explored. To this effect, experimental studies assessing the impact of IH and SF in MR transgenic animals with vascular-specific deletion of MR are essential to elucidate the role of MR in OSA-induced CVD. Furthermore, the use of MR antagonists in OSA in the context of clinical trials in focused experimental settings may also provide insights into the effectiveness of MR antagonism in mitigating OSA-mediated CVD. Therefore, MR could prove to be a valuable putative target for improving cardiovascular outcomes in OSA patients.

## Figures and Tables

**Figure 1 ijms-24-02245-f001:**
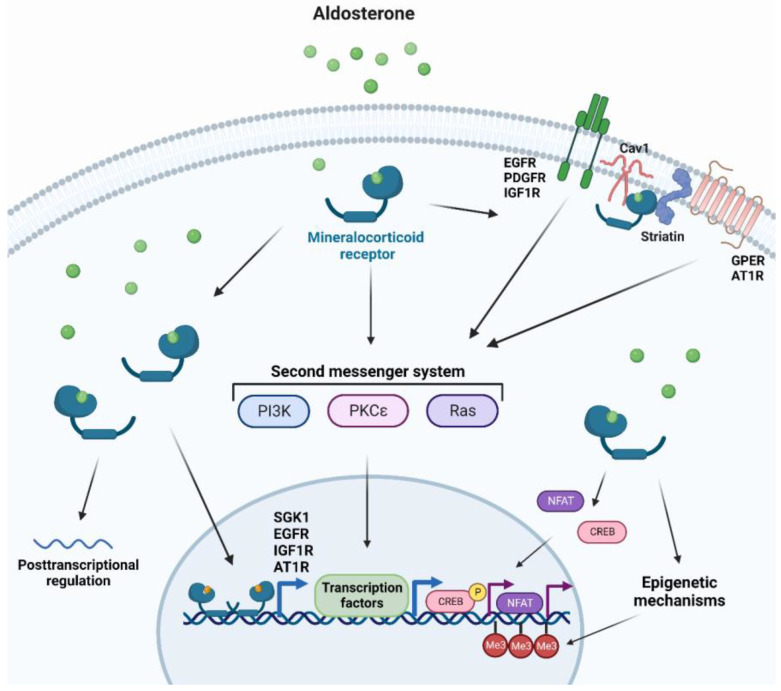
**Cellular responses to MR activation.** Aldosterone binding to cytosolic mineralocorticoid receptors forms homodimers and translocates into the nucleus to elicit genomic responses. Cytosolic MR activation can also trigger non-genomic rapid responses through second messenger systems. Additionally, MR activation can transactivate several membrane-bound receptors, crosstalk with other cytosolic pathways, induce or modify epigenetic modifications, and affect posttranslational regulation. AT1R: Type I angiotensin receptor, Cav1: caveolin 1, CREB: cAMP-response element-binding protein, EGFR: epidermal growth factor, GPER: G protein-coupled estrogen receptor 1, IGF1R: insulin-like growth factor receptor 1, Me3: tri-methylation, NFAT: Nuclear factor of activated T-cells, PDGFR: platelet-derived growth factor receptor, PI3K: phosphatidylinositide 3-kinases, PKC: protein kinase C, SGK1: serine/threonine-protein kinase 1.

**Figure 2 ijms-24-02245-f002:**
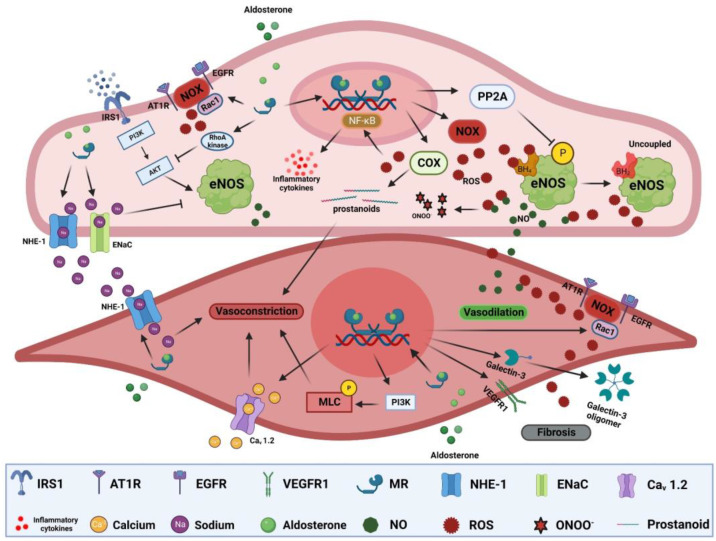
**MR-mediated mechanisms of vascular dysfunction.** MR activation in endothelial cells can induce ROS production through activation of NOXs and COXs. ROS react with NO, reducing its bioavailability leading to impaired endothelial-dependent vasodilation, forming a more reactive nitrogen species, ONOO-, oxidizing BH_4_ to uncouple eNOS that leads to production of ROS instead of NO. Excessive ROS production can also trigger NF-ƙB activation, leading to increased expression of inflammatory cytokines. MR activation can also inhibit eNOS through dephosphorylation by PP2A, impair insulin-dependent signaling, activate NHE-1 and EnaC. In SMCs, MR activation can trigger vasoconstriction through activation of NHE-1 and Ca_v_ 1.2 in addition to MLC phosphorylation. Activation of MR can elicit fibrotic signaling through increased ROS production, galectin-3 oligomer formation, and VEGEF signaling. Akt: protein kinase B, AT1R: type I angiotensin receptor, BH_4_: tetrahydrobiopterin, Ca_v_ 1.2: L-type calcium channel alpha 1C, COX2: cyclooxygenase 2, EGFR: epidermal growth factor, EnaC: epithelial sodium channel, eNOS: endothelial nitric oxide synthase, IRS1: insulin receptor substrate 1, MLC: myosin light chain, NHE-1: sodium–hydrogen antiporter 1, NF-ƙB: nuclear factor kappa B, NOX: NADPH oxidases, ONOO^-^: peroxynitrite, PI3K: phosphatidylinositide 3-kinases, PP2A: protein phosphatase 2A, Rac1: Ras-related C3 botulinum toxin substrate, ROS: reactive oxygen species, VEGFR1: vascular endothelial growth factor receptor 1.

**Figure 3 ijms-24-02245-f003:**
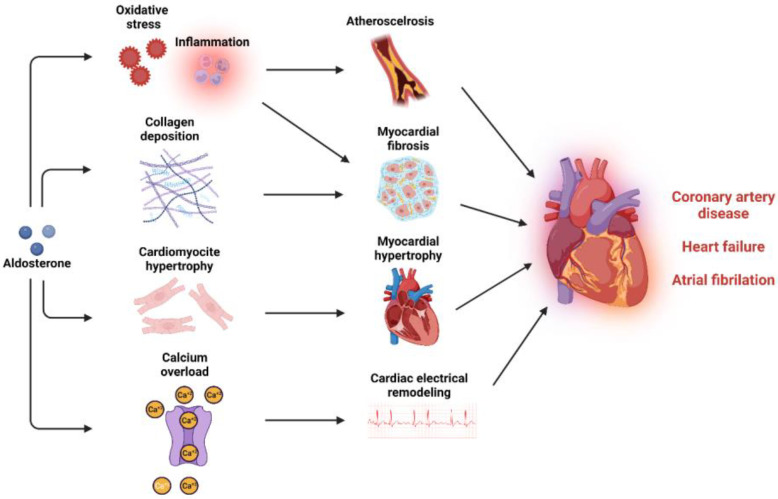
**Mineralocorticoid activation and cardiovascular disease.** MR activation can induce multiple pathological mechanisms that promote atherosclerosis and myocardial dysfunction and can lead to coronary artery disease, heart failure, and atrial fibrillation.

**Figure 4 ijms-24-02245-f004:**
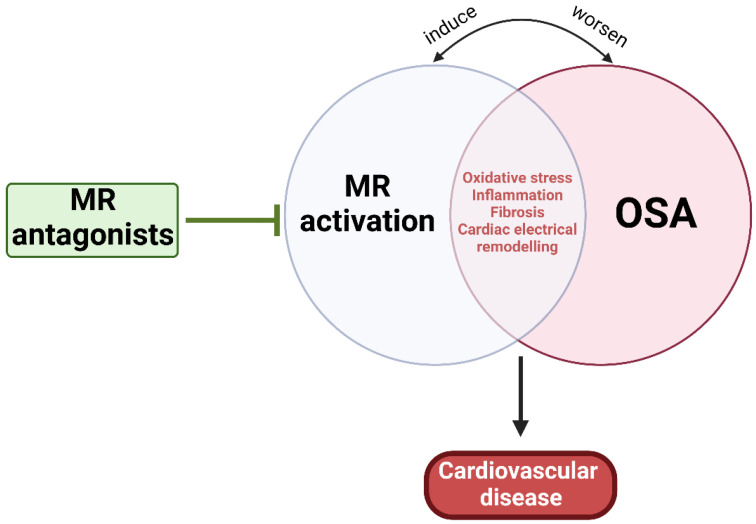
**Pathophysiological relationship between OSA and MR activation.** MR activation and OSA share multiple mechanisms that can lead to CVD, suggesting the potential benefit of MR antagonism.

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
