# Peer review of "Cardiovascular Disease in Obstructive Sleep Apnea: Putative Contributions of Mineralocorticoid Receptors"

_ijms, 2023, doi:10.3390/ijms24032245_

Round 1
Reviewer 1 Report
Badran, et al. reviewed the role of mineralocorticoid receptors in cardiovascular disease risk associated with obstructive sleep apnea. The manuscript is a comprehensive survey of the literature and will be a useful resource for others in the field. However, I have some suggestions to improve the manuscript:
Major comments
· Reorganizing some parts of the manuscript would improve the flow and tighten up the text. The section discussing the cardiovascular outcomes of OSA (pages 17-19) should be moved to the beginning of the manuscript, either incorporated into the introduction or as a separate section. Also, the discussion of CPAP on pages 21-22 should be moved to the beginning of the manuscript. This would strongly emphasize the need for additional treatments aimed at improving CV outcomes in OSA patients and set the stage for the discussion of MR activation as a potential therapeutic target.
· The section entitled “MR-mediated vascular dysfunction” should be combined with the prior section “Pathological mechanisms of MR activation.” The section on endothelial dysfunction could follow the discussion of oxidative stress and the section on arterial stiffness could follow the discussion of fibrosis.
· Figure 2 is very complex and some of the symbols look similar, making it difficult to interpret. For example, both aldosterone and nitric oxide seem to be represented by green circles, and it would be helpful if the symbols were more distinct. Adding in a symbol legend would improve interpretability, as well as help make the figure less crowded. In addition, a comprehensive figure legend should be added so that the figure stands on its own separate from the text.
Minor comments
· Page 7 – typo in spelling of 8-isoprostane (8-isprostane)
· The presentation of trial names vs. acronyms should be consistent throughout the manuscript. Sometimes the trial name is given followed by the acronym in parentheses and sometimes it is reversed (acronym followed by trial name in parentheses). Also, on page 12 the acronym should be “CHARM” rather than “CAHRM.”
Author Response
Open Review #1
Comments and Suggestions for Authors
Badran, et al. reviewed the role of mineralocorticoid receptors in cardiovascular disease risk associated with obstructive sleep apnea. The manuscript is a comprehensive survey of the literature and will be a useful resource for others in the field. However, I have some suggestions to improve the manuscript:
Major comments
- Reorganizing some parts of the manuscript would improve the flow and tighten up the text. The section discussing the cardiovascular outcomes of OSA (pages 17-19) should be moved to the beginning of the manuscript, either incorporated into the introduction or as a separate section. Also, the discussion of CPAP on pages 21-22 should be moved to the beginning of the manuscript. This would strongly emphasize the need for additional treatments aimed at improving CV outcomes in OSA patients and set the stage for the discussion of MR activation as a potential therapeutic target.
- The section entitled “MR-mediated vascular dysfunction” should be combined with the prior section “Pathological mechanisms of MR activation.” The section on endothelial dysfunction could follow the discussion of oxidative stress and the section on arterial stiffness could follow the discussion of fibrosis.
Thank you for your suggestions. We now reorganized the parts according to your recommendations.
- Figure 2 is very complex and some of the symbols look similar, making it difficult to interpret. For example, both aldosterone and nitric oxide seem to be represented by green circles, and it would be helpful if the symbols were more distinct. Adding in a symbol legend would improve interpretability, as well as help make the figure less crowded. In addition, a comprehensive figure legend should be added so that the figure stands on its own separate from the text.
Thank you. We added a symbol legend and comprehensive figure legends for all figures.
Minor comments
- Page 7 – typo in spelling of 8-isoprostane (8-isprostane)
Fixed. Thank you.
- The presentation of trial names vs. acronyms should be consistent throughout the manuscript. Sometimes the trial name is given followed by the acronym in parentheses and sometimes it is reversed (acronym followed by trial name in parentheses). Also, on page 12 the acronym should be “CHARM” rather than “CAHRM.”
The presentation of trial names vs. acronyms are now consistent. Thank you for pointing that out.
Reviewer 2 Report
The authors of the article entitled “Cardiovascular Disease in Obstructive Sleep Apnea: Putative Contributions of Mineralocorticoid Receptors” have described an important roles of mineralocorticoid receptors (MR) in cardiovascular disease and obstructive sleep apnea (OSA). They have reviewed in detail the function and activation of MR and MR-mediated vascular insufficiency and cardiovascular disease pathogenesis. In addition, they have showed that OSA is an independent risk factor for various cardiovascular diseases.
Apnea and activation of the sympathetic nervous system (SNS) are also important, as reported by Floras et al (Hypertension. 2011 Sep;58(3):e17-8. doi: 10.1161/HYPERTENSIONAHA.111.177907.). It would be worthwhile discussing the association between SNS and MR activation should be mentioned.
Author Response
Open Review #2
Comments and Suggestions for Authors
The authors of the article entitled “Cardiovascular Disease in Obstructive Sleep Apnea: Putative Contributions of Mineralocorticoid Receptors” have described an important roles of mineralocorticoid receptors (MR) in cardiovascular disease and obstructive sleep apnea (OSA). They have reviewed in detail the function and activation of MR and MR-mediated vascular insufficiency and cardiovascular disease pathogenesis. In addition, they have showed that OSA is an independent risk factor for various cardiovascular diseases.
Apnea and activation of the sympathetic nervous system (SNS) are also important, as reported by Floras et al (Hypertension. 2011 Sep;58(3):e17-8. doi: 10.1161/HYPERTENSIONAHA.111.177907.). It would be worthwhile discussing the association between SNS and MR activation should be mentioned.
Thank you for your suggestion. We added a paragraph on MR activation and sympathetic activity in page 12.
Round 2
Reviewer 1 Report
I appreciate the time and effort the authors took in addressing my comments. I have no further suggestions.